# An open source three-mirror laser scanning holographic two-photon lithography system

**Marco Pisanello[1]\*, Di Zheng[1], Antonio Balena[1], Filippo Pisano [1], Massimo De Vittorio[1,2]‡\*, Ferruccio Pisanello [1]‡\***

1 Center for Biomolecular Nanotechnologies – Istituto Italiano di Tecnologia, Arnesano (LE), Italy,
2 Dipartimento di Ingegneria dell'Innovazione – Università del Salento, Lecce (LE), Italy

☉ These authors contributed equally to this work.
‡ MDV and FP also contributed equally to this work.
\* marco.pisanello@iit.it (MP); massimo.devittorio@iit.it (MDV); ferruccio.pisanello@iit.it (FP)

**Data Availability Statement:** All relevant data are within the article and its Supporting information files.

## Abstract

Two-photon polymerization is a widely adopted technique for direct fabrication of 3D and 2D structures with sub-diffraction-limit features. Here we present an open-hardware, open-software custom design for a holographic multibeam two-photon polymerization system based on a phase-only spatial light modulator and a three-mirror scanhead. The use of three reflective surfaces, two of which scanning the phase-modulated image along the same axis, allows to overcome the loss of virtual conjugation within the large galvanometric mirrors pair needed to accommodate the holographic projection. This extends the writing field of view among which the hologram can be employed for multi-beam two-photon polymerization by a factor of ~2 on one axis (i.e. from ~200μm to ~400μm), with a voxel size of ~250nm × ~1050nm (lateral × axial size), and writing speed of three simultaneous beams of 2000 voxels/s, making our system a powerful and reliable tool for advanced micro and nano-fabrications on large area.

## 1. Introduction

In the last two decades the use of additive manufacturing techniques to realize two- and three-dimensional structures at the micro- and nano-scale has been widely employed to obtain high resolution structures [1, 2]. Particularly, techniques based on Two-Photon Polymerization (2PP) of photoresists emerged by exploiting their ability to achieve direct writing of sub-diffraction limited structures in three dimensions, with limited constraints on their shapes, and in an extensive set of different materials, including acrylate- and epoxy-based resins, hydrogels, silicate-based photopolymer [1, 3–5].

The typical 2PP fabrication system allowing for sub-micrometer resolution employs a focused fs-pulsed near infrared laser spot scanned in a writing field of view (WFOV) ~0.020 mm$^2$ (~140 × 140 μm$^2$) [3] square frame by means of galvanometric mirrors and a scan-tube lenses relay system. By translating either the substrate or the objective perpendicularly to the scanning plane, 3D polymeric structures can be realized with sub-diffraction limited features [3] that can extend even over a few millimeters from the substrate [1].

**Funding:** F.Pisano, A.Balena, and F.Pisanello acknowledge funding from the European Research Council under the European Union's Horizon 2020 Research and Innovation Program (Grant Agreement No. 677683). D.Zheng, M.De Vittorio, and F.Pisanello acknowledge funding from the European Union's Horizon 2020 Research and Innovation Program (Grant Agreement No. 828972). F.Pisano, M.De Vittorio, and F.Pisanello acknowledge that this project has received funding from the European Union's Horizon 2020 Research and Innovation Program (Grant Agreement No 101016787). M.Pisanello and M.De Vittorio acknowledge funding from the European Research Council under the European Union's Horizon 2020 Research and Innovation Program (Grant Agreement No. 692943). M.De Vittorio and F. Pisanello acknowledge that this project has received funding from the European Union's Horizon 2020 Research and Innovation Program (Grant Agreement No. 966674). M.Pisanello, F. Pisanello, and M.De Vittorio were funded by the U. S. National Institutes of Health (Grant No. 1UF1NS108177-01). The funders had no role in study design, data collection and analysis, decision to publish, or preparation of the manuscript.

**Competing interests:** I have read the journal's policy and the authors of this manuscript have the following competing interests: M.D.V. and F. Pisanello are founders and hold private equity in OptogeniX srl, a company that develops, produces and sells technologies to deliver light into the brain. This does not alter our adherence to PLOS ONE policies on sharing data and materials. OptogeniX did not fund the research described in this work. M.D.V.: OptogeniX srl (I). F.P.: OptogeniX srl (I).

However, the use of a single writing spot can lead to long fabrication times; Malinauskas *et al.* [6] provide a very simple formula to estimate the time *t* needed to fabricate a specific structure based on its width, length, and height (*w*, *l*, and *h*, respectively) and on the cross-section area *R* of the polymerizing voxel, fill factor *F*, and sample scanning speed *v*:

$$t = \frac{w \cdot l \cdot h \cdot F}{R \cdot v}. \tag{1}$$

As an example, an $800 \times 800 \times 40 \ \mu m^3$ structure with $F = 0.5$ written at 1mm/s with a 1.25NA objective results in *t* approaching 7 hours [6]. To overtake this issue, together with chemical engineering aimed at improving photoresists performances in terms of two-photon absorption cross-section and polymerization 2rate [1, 3], holographic techniques allowed direct shaping of the polymerized structures. This has been achieved by direct projection of complex shaped beams as well as multi-spot writing, both with static phase masks or microlens arrays [7–9] and reconfigurable holograms obtained by dynamic optoelectronic elements to modulate phase and/or intensity of the light field. These latter include Spatial Light Modulators (SLMs) or Digital Micromirror Devices (DMDs), with the use of dynamic phase masks allowing for engineered beam shape both in the transversal plane[10–13] and the axial direction[12, 14, 15].

The simultaneous use of phase modulating elements and galvanometric scanning solutions, however, represents a challenge that requires the use of larger mirrors with respect to single-beam raster-scanning systems, in order to accommodate larger and shaped beams. As a consequence, the distance between the two virtually conjugated mirrors increases, affecting the validity of the virtual conjugation approximation. This ultimately results in severe vignetting of the effective field of view, an undesired drawback for the system's throughput.

In this work we present a custom designed holographic 2PP system based on a SLM and a three-mirror scan head. Two out of the three mirrors deflect the laser light along the same axis (*x* axis hereinafter) to counteract the beam displacement on the surface of the *y* axis mirror, with minimal additional costs and relatively easy implementation. This maintains the projection of the SLM plane stationary on the entrance plane of the scan lens/tube lens system. By implementing this solution the WFOV along *x* can be improved with respect to a typical two-mirror 2PP holographic approach [1, 11, 16], making the system more inclined toward holographic large-scale fabrications. Indeed, the virtual conjugation is reinforced, and the distortions at the edges of a relatively large WFOV of $\sim 400 \times 400 \ \mu m^2$ ($\sim 0.16 \ mm^2$) can be reduced. We provide a design strategy fully based on off-the-shelf hardware from well-known suppliers, to avoid the use of expensive custom-made optics, optomechanics, and electronic equipment, and to ease replication of the system. A full scheme of the system is available as a three dimensional Computer-Aided Design (CAD) model as S1 File, with a list of all the elements given in S2 File. As a general guideline, we also provide the source code of the custom-written MATLAB software developed to control the system as S3 File.

## 2. Results

### 2.1 System overview and design concepts

The layout of the optical system is depicted in Fig 1, in which four main blocks can be identified: *(I)* the power control path, acting as a fast shutter and managing the fine tuning of the light power entering the system via an electro-optic modulator (Conoptics, 350-80-02), *(II)* the holography path, impressing a phase modulation on the laser beam wavefront by means of a SLM, *(III)* the scanning path, translating the hologram reconstruction within the WFOV, and *(IV)* the imaging path, allowing for a real time monitoring of the process. Each of the blocks

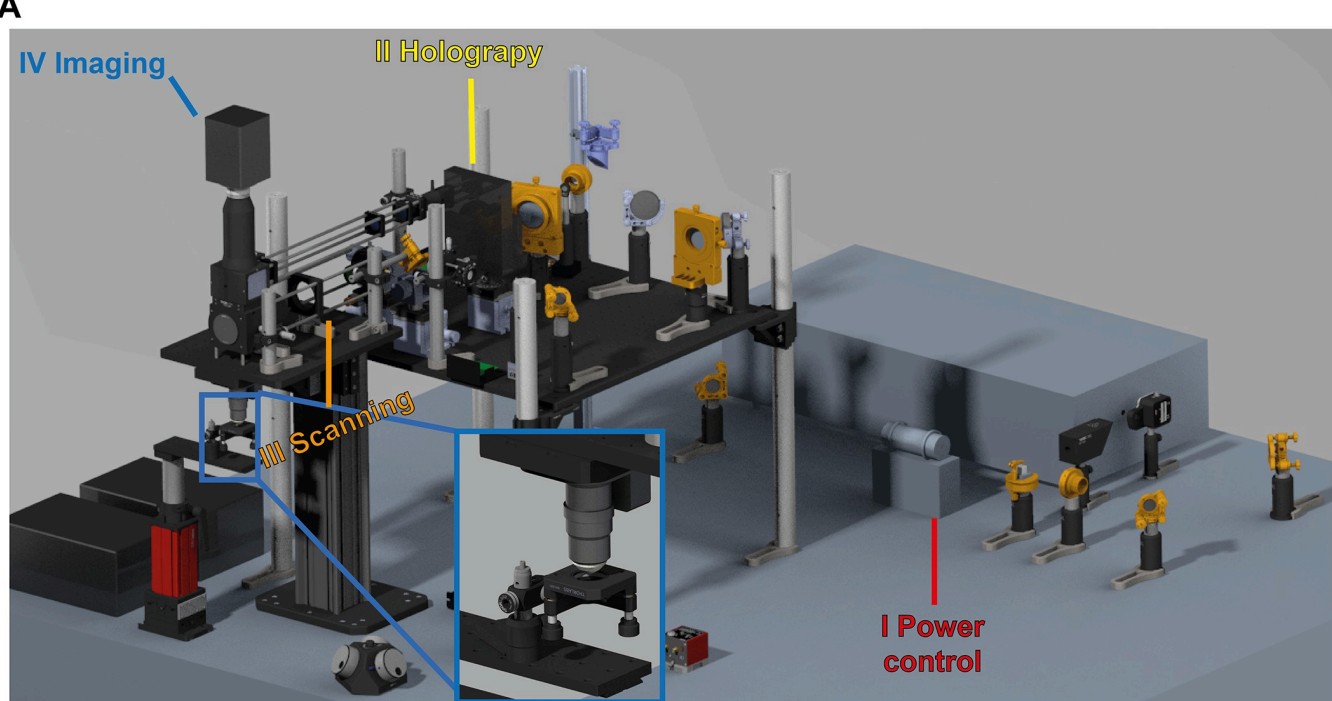

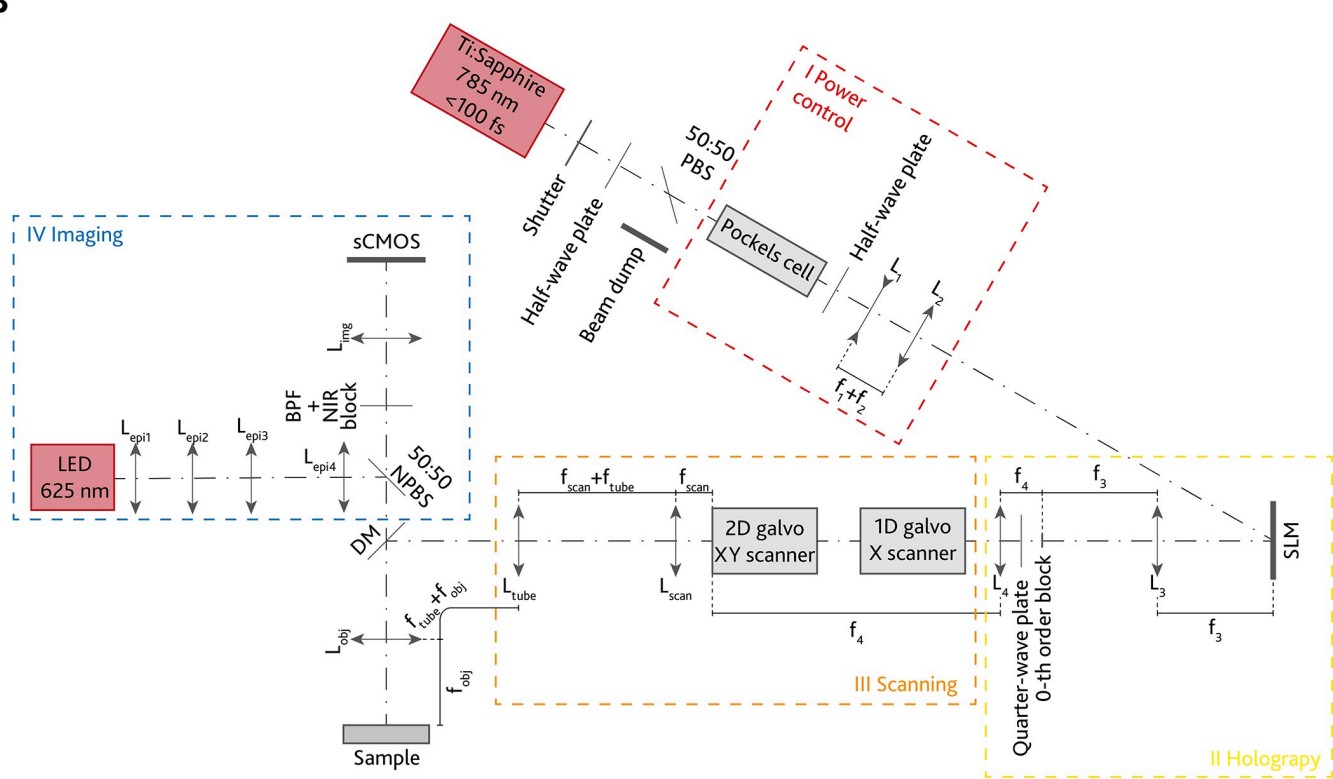

**Fig 1. Optical scheme of the system.** (A) Overall view from the Computer-Aided Design (CAD) model. The inset shows a magnification of the sample holder and the objective lens. (B) Schematic representation of the system. After power control (I), a fs-pulsed laser beam is phase modulated by the SLM (II) and the reconstructed hologram can be translated within the field of view through a virtually conjugated galvo triplet (III). An epi-illumination and imaging system (IV) allow for real time monitor of the lithographic process. Details on all the optical and optomechanical elements are given in S2 File.

*(I)-(III)* provide a beam magnification $M_I$, $M_{II}$ and $M_{III}$ constrained by the apertures of the following optical elements on the path. Given a beam diameter $d_0$ at the laser output, $M_I$ should allow to overfill the SLM chip, $M_{II}$ should get the beam from the SLM chip and resize it to fit the entrance pupil $A_s$ of block *(III)*, while the pair scan-tube lens in block *(III)* should magnify the beam of a factor $M_{III}$ to slightly overfill the objective's back aperture $A_{OBJ}$. This results in the following design rules:

$$\begin{cases} M_I > \dfrac{\max(\{v_{SLM}, h_{SLM}\})}{d_0} \\[2mm] M_{II} < \dfrac{A_S}{d_{SLM}} \\[2mm] M_{III} \cdot M_{II} < \dfrac{A_{OBJ}}{d_{SLM}} \end{cases} \qquad (2)$$

where $v_{SLM}$ and $h_{SLM}$ are the lengths of the SLM chip sides. Polarization control is performed along the optical path by half-wave and quarter-wave plates (10RP52-2B, Newport and 10RP54-2B, Newport, respectively) to maximize the SLM modulation efficiency and to enter the scan path with circularly polarized light. To accommodate the wide tunability of the laser source in the 690nm-1050nm range (Coherent Chameleon Vision-S), all optical elements were chosen with anti-reflective coating and achromatic performance compatible with this wavelength interval. All experiments reported in this manuscript were performed with the laser tuned to 785nm.

Being the objective lens and the SLM the key parts of the system, since they affect the most both phase modulation and resolution performances, the design was carried out around these two components. The objective lens should provide a theoretical lateral dimension of the spot $r_{xy} < 500$ nm at $\lambda = 785$ nm, and a working distance of several millimeters to allow the realization of tall structures. Simultaneously, a high-resolution phase modulator is needed to maximize the beam displacement that can be impressed holographically [17]. To avoid bulky and expensive custom optics, high-end off-the-shelf components were chosen. Olympus XLPLN25XWMP2 matches the requirements on the objective lens, providing a numerical aperture NA = 1.05 ($r_{xy} \sim 345$ nm at $\lambda = 785$ nm [18]), with a working distance of 2 mm and entrance pupil diameter $A_{OBJ}\sim15$mm. To allow for fast-changing phase modulation, a 712 Hz refresh rate SLM with $1920 \times 1152$ pixels at a pitch $p_{SLM} = 9.2$ μm (Meadowlark Optics, HSP1920-500-1200-HSP8-785, chip size $h_{SLM} = 17.6$ mm $\times v_{SLM} = 10.7$ mm) was selected. According to Eq (1), matching the beam size to both the entrance pupil of the objective and the shorter side of the SLM requires $M_{III} \cdot M_{II} \approx 1.35$.

To extend the angular range in which the virtual conjugation of the $G_x$, $G_y$ galvanometer pair holds, the scanhead is augmented by a third element $G_{x2}$ counteracting the beam movement on $G_y$: by having the first two galvanometers on the path ($G_{x2}$ and $G_x$) deflecting the beam along $x$, the input beam of the scan and tube relay can be maintained stationary on the surface of $G_y$, opposing the effect of the increased distance between the mirror due to their size.

## 2.2. Holography path

The holography path (block II) impresses a phase modulation on the laser beam wavefront with the goal to reproduce a certain target image on the sample plane. The SLM plane is therefore optically conjugated to the back aperture of the objective by means of two achromatic relay telescopes: one is formed by lenses $L_3$ and $L_4$, conjugating the SLM plane with the third galvanometric mirror in the scan path, while the other is formed by the scan and tube lenses ($L_{scan}$, $L_{tube}$), conjugating the third mirror to the back aperture of the objective. In such

configuration, the maximum deflection that can be impressed by the SLM in the sample plane is given by [19]

$$\Delta x_{\max} = \frac{f_{\mathrm{OBJ}} \cdot \lambda}{2 \cdot M_{\mathrm{III}} \cdot M_{\mathrm{II}} \cdot p_{\mathrm{SLM}}},$$ (3)

where $f_{\mathrm{OBJ}}$ is the effective focal length of $L_{\mathrm{OBJ}}$. To reasonably match the entrance pupil of the objective given $M_{\mathrm{III}} = 4$ (according to the details on scanning optics in the section 2.3), lenses $L_3$ and $L_4$ were chosen with focal lengths $f_3 = 500$ mm and $f_4 = 160$ mm (Edmund Optics 49–396 and #67–334, respectively), giving $M_{\mathrm{II}} = 0.32$ and a theoretical Hologram Field Of View HFOV $= 2 \cdot \Delta x_{\max} \approx 480$ μm at the sample plane. The unmodulated $0^{\mathrm{th}}$ order diffracted from the SLM was physically blocked with a razorblade placed in the focal plane of $L_3$.

## 2.3 Scanning path

The phase modulated beam is sent into a virtually conjugated galvo triplet (Fig 2A and 2B), in which the first two mirrors ($G_{x2}$, $G_x$) scan the laser along the same axis so that the beam remains stationary on the surface of the third mirror ($G_y$), this latter scanning the beam along the other axis [20]. By doing so, the distortions that would have been introduced by the virtually conjugated galvo pair alone due to the beam drifting on the surface of $G_y$ are mitigated. A MATLAB code based on ray optics consideration (S4 File) was used to compute an estimation of the relation between the tilts of mirrors $G_{x2}$ and $G_x$, given the distances between the different components of the scan head $l$ and $d$, defined in Fig 2A. The scan and tube lenses (Thorlabs SL50-CLS2 and TL200-CLS2, respectively), chosen with focal lengths suited to match commercially available objectives from different manufacturers, give a magnification $M_{\mathrm{III}} = 4$. Since $A_{\mathrm{OBJ}} \sim 15$mm, the beam size at the entrance of the scan tube should be ~4mm ($A_{\mathrm{OBJ}}/M_{\mathrm{III}} = 3.75$) to slightly overfill the objective back aperture, and galvanometer mirrors suited for 10 mm beams were chosen (Thorlabs GVS011/M and GVS012/M) to easily accommodate the beam on the surface of all the mirrors. Fig 2C shows a comparison of the beam displacement on the surface of $G_y$ with and without $G_{x2}$, when forcing a direct proportionality between the tilt of $G_{x2}$ and $G_x$ with a factor -1.4, and with l = 90mm d = 14.7mm.

To allow for a fast and accurate 3D scan, the objective lens was mounted on a piezoelectric focuser (PI PIFOC P-725.4CD).

## 2.4. Imaging path

To allow for real time monitoring of the 2PP process, the imaging path consists in an epi-illumination arm equipped with a 625 nm LED source (M625L4, Thorlabs) for sample illumination and a sCMOS camera (Orca Fusion, Hamamatsu). A dual bandpass filter (BPF, FF01-512/630-25, Semrock), together with a NIR blocker filter (FF01-680/SP-25, Semrock), permits imaging at both LED and polymer fluorescence wavelengths. The image is projected on the sCMOS camera through a Thorlabs TTL200 tube lens ($f_{\mathrm{img}} = 200$mm). The epi-illuminator consists of four lenses indicated with $l_{\mathrm{epi1-4}}$ in Fig 1, being Thorlabs AC254-030-A ($f_{\mathrm{epi1}} = 30$mm), LBF254-040-A ($f_{\mathrm{epi2}} = 40$mm), LBF254-040-A ($f_{\mathrm{epi3}} = 40$mm), and AC254-250-A ($f_{\mathrm{epi4}} = 250$mm) respectively, arranged (together with the objective) in a Kohler illumination scheme.

## 2.5. Software/Hardware interface

A custom MATLAB software has been developed to control the system, with the source code available in S3 File. Data acquisition boards from National Instruments (NI PXIe-1073

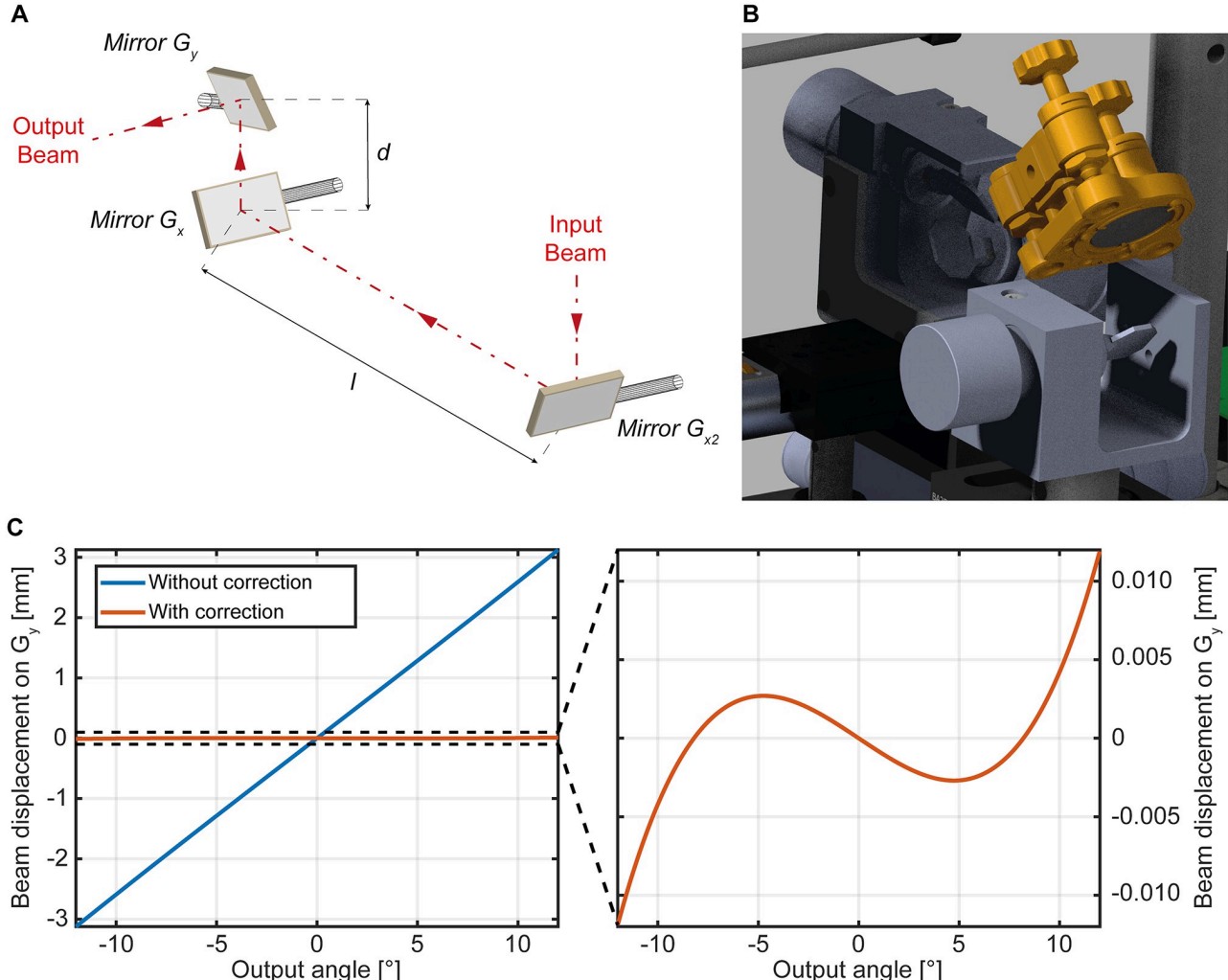

**Fig 2. The three-mirror scanhead.** (A) Schematic representation of the virtually conjugated galvo triplet. To mitigate the distortion introduced by the beam being not stationary in a virtually conjugated galvo couple, a scan system with three galvanometric mirrors placed at mutual distance l and d is implemented. (B) Detail of the three-mirror scanhead from the CAD design. (C) Comparison between the beam displacement on the surface of Gy whit and without the use of Gx2.

equipped with NI PXIe-6363) operating at 512 ksamples/s are employed as software/hardware interface. In addition to timing control of laser power and mirrors position during fabrication, a routine to assist in finding the position of the photoresist-coverslip interface has been implemented by reading the fluorescence intensity generated within the photoresin during a scan along the axis of the objective, with laser power set below the polymerization threshold.

For hologram generation, the phase patterns fed to the SLM were computed through the Gerchberg–Saxton algorithm [21] or the Mixed Region Amplitude Freedom (MRAF) [22] on 1920 × 1920 pixels images. The algorithm ran for 100 iterations.

## 2.6 System characterization

The performances of the system were first evaluated with the unmodulated 0[th] order diffracted by the SLM. A fluorescent droplet (30 μM fluorescein:PBS solution) was placed on a coverslip and the laser beam scanned with the galvanometric triplet. Fig 3A shows a 9 × 9 fluorescence

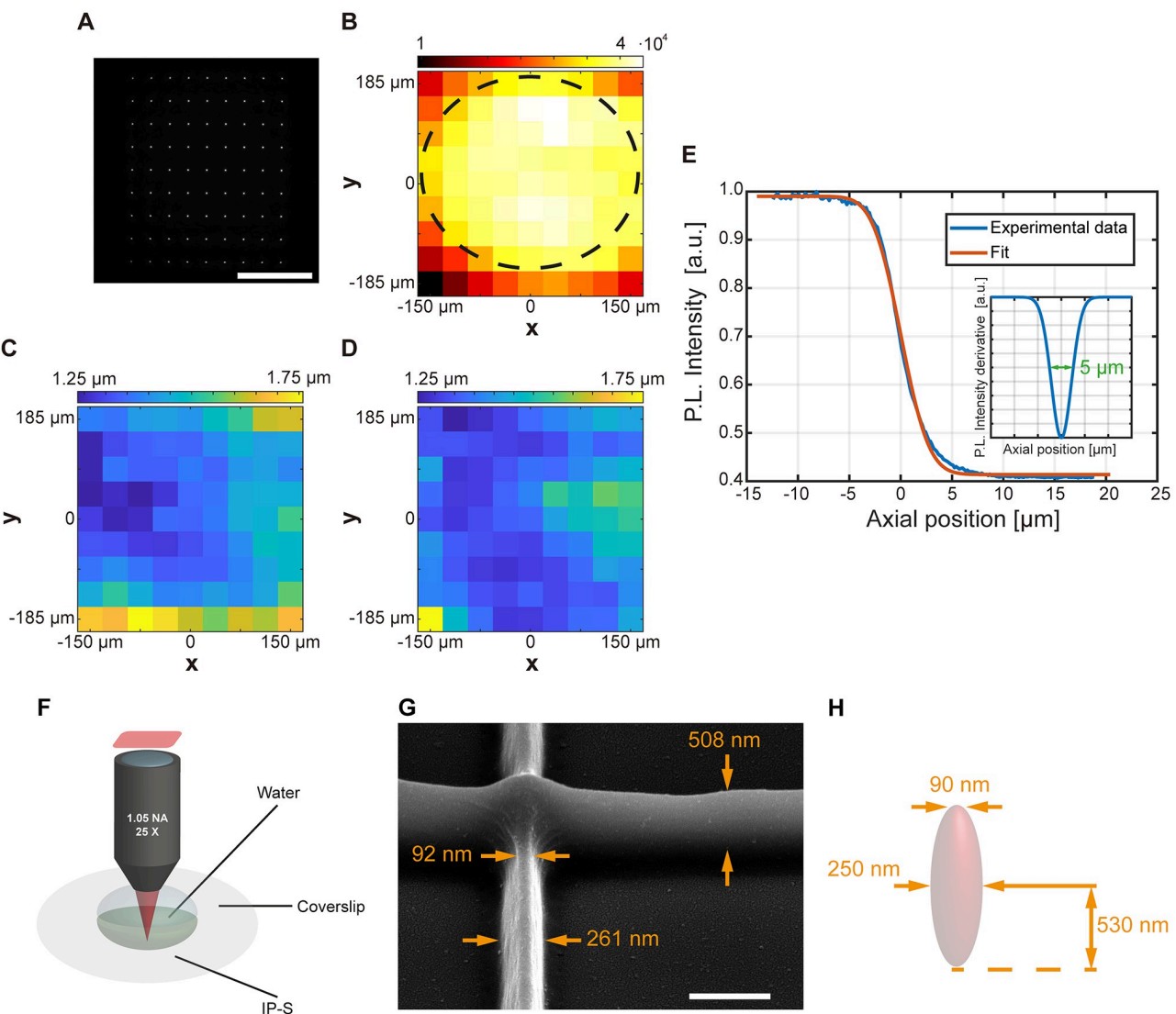

**Fig 3. Spot characterization.** (A) Matrix of fluorescence spots with 785 nm femtosecond laser excitation. Scalebar is 150μm (B) Maximum intensity maps of the fluorescence spots in (A). The black dashed line highlights the region with standard deviation less than 5%. (C) x dimension map of the fluorescence spots in (A). (D) y dimension map of the fluorescence spots in (A). (E) Estimation of the axial size of the spot. (F) Water-coverslip-photoresist configuration used for structure fabrication with the water immersion objective. (G) Details of the minimum line thickness of ~90nm achieved with laser power 14.1mW at the sample plane (estimated power density of 7.18 TW/cm$^2$ [23]) and writing speed 0.01mm/s. Scalebar is 500nm; sample is tilted by 45˚. (H) Schematic representation of the polymerizing voxel. Each dimension is obtained as average on n = 5 measurements, standard deviation is 5% for voxel diameters (both central and superficial) and 4% for voxel half-height.

spots matrix excited in the fluorescent drop, recorded by scanning the focused laser beam while exposing the sCMOS camera for a time long enough to capture all the matrix elements. For each focused spot, both intensity and beam size were evaluated. As shown by the color map in Fig 3B, beam intensity has a standard deviation below 5% in a circular region of diameter 300 μm (outlined by the black dashed), that increases to <15% if a square 300μm x 300μm field is considered. The size of each fluorescence spot was then analyzed by extracting the intensity profile along the *x* and *y* axes to compute the related full width at half maximum (FWHM) distribution, reported in Fig 3C and 3D. Across the 300μm x 300μm field, average FWHM was estimated to be 1.36μm with a standard deviation 0.09 μm, on a total of n = 144

measurements (72 points per axis). An axial dimension of 5μm was estimated as the FWHM of the derivative of the fluorescent signal acquired while scanning the spot from the fluorescein drop to the glass slide at the center of the WFOV (Fig 3E).

To evaluate the minimum feature size polymerizable by the system, set of lines were written by using the unmodulated $0^{th}$ order in a water/coverslip/IP-S photoresist (Nanoscribe GmbH) configuration (Fig 3F). By tuning laser power and writing speed of continuous structures, a minimum line thickness of ~90 nm was obtained, as shown in Fig 3G. These continuous structures were used to estimate the minimum voxel size, reported in Fig 3H, on the basis of the average among n = 5 measurements per dimension.

### 2.7 Holographic 2PP with a three-mirror scan head

By activating the SLM phase modulation, the system allows projecting holographic patterns as shown by multiple spots and more complex shapes generated in a fluorescence drop (Fig 4A and 4B). The projection can be reliably translated by the galvanometric mirror triplet in the $xy$ plane, covering a field of view that can range up to about 400 μm × 400 μm (Fig 4B). As the galvanometric triplet is aimed at increasing the writing field of view of holographic 2PP along $x$, we have tested the extension of polymerized structures realized by activating or not the mirror $G_{x2}$. A three-spot hologram (Fig 4C) was projected into the photoresist and scanned along $x$ in both discrete or continuous fashion to realize arrays of polymerized pillars or lines, respectively. Comparison of the obtained patterns are reported in Fig 4D and 4E with $G_{x2}$ ON and OFF, respectively. For both continuous and discrete scanning by activating $G_{x2}$ the WFOV along $x$ almost doubles, extending from ~200μm to ~400μm. Fig 4D and 4E also report high magnification SEM images with detailed views of pillars and lines with laser power 32.6mW (dwell time 200ms for pillars and writing speed 0.02mm/s for lines, estimated power density of 5.53 TW/cm$^2$ per spot [23]). At the center of the WFOV the feature size remains almost unchanged between the two conditions $G_{x2}$ ON or OFF, while the smaller WFOV when $G_{x2}$ is not enabled make the structures fading at about 100μm from the center of the WFOV. This effect is slightly visible also when $G_{x2}$ is active, despite happening ~200μm away from the WFOV center.

The same patterns were written by selecting different laser powers and dwell times/writing speeds, obtaining the structure shown in Fig 5A–5D. While the structure thickness remained in the order of hundreds of nanometers, spanning from 300nm to 900nm (as shown in Fig 5C and 5D, respectively), the structure height varied from ~300 nm to ~4 μm (as shown in Fig 5B and 5C, respectively), depending on the writing parameters. Hence, by tuning the writing parameters the form factor of the realized structures ranged from 1 to about 12. Furthermore, as shown in Fig 6, the system was used to polymerize simple 3D structures in the shape of pyramids. As the memory buffer of the system is limited, the sample rate of the hardware/software interface was reduced to 32ksamples/s in this case.

## 3. Discussions

In this work we describe the implementation of an open source three-mirror scanhead system to increase the writing field of view of holographic two-photon photopolymerization (2PP) systems. The approach is fully realized with off-the-shelf components, and the additional galvanometric mirror is employed to reduce the beam drift on the last reflective element of the scanhead. As beam walking compensation has been already employed in fluorescence microscopes, we take advantage of this method to circumvent the loss of virtual conjugation due to increased galvanometric mirrors size to host the holographic projection. Indeed, in holographic systems the laser beam is expanded to slightly overfill the modulating element, and

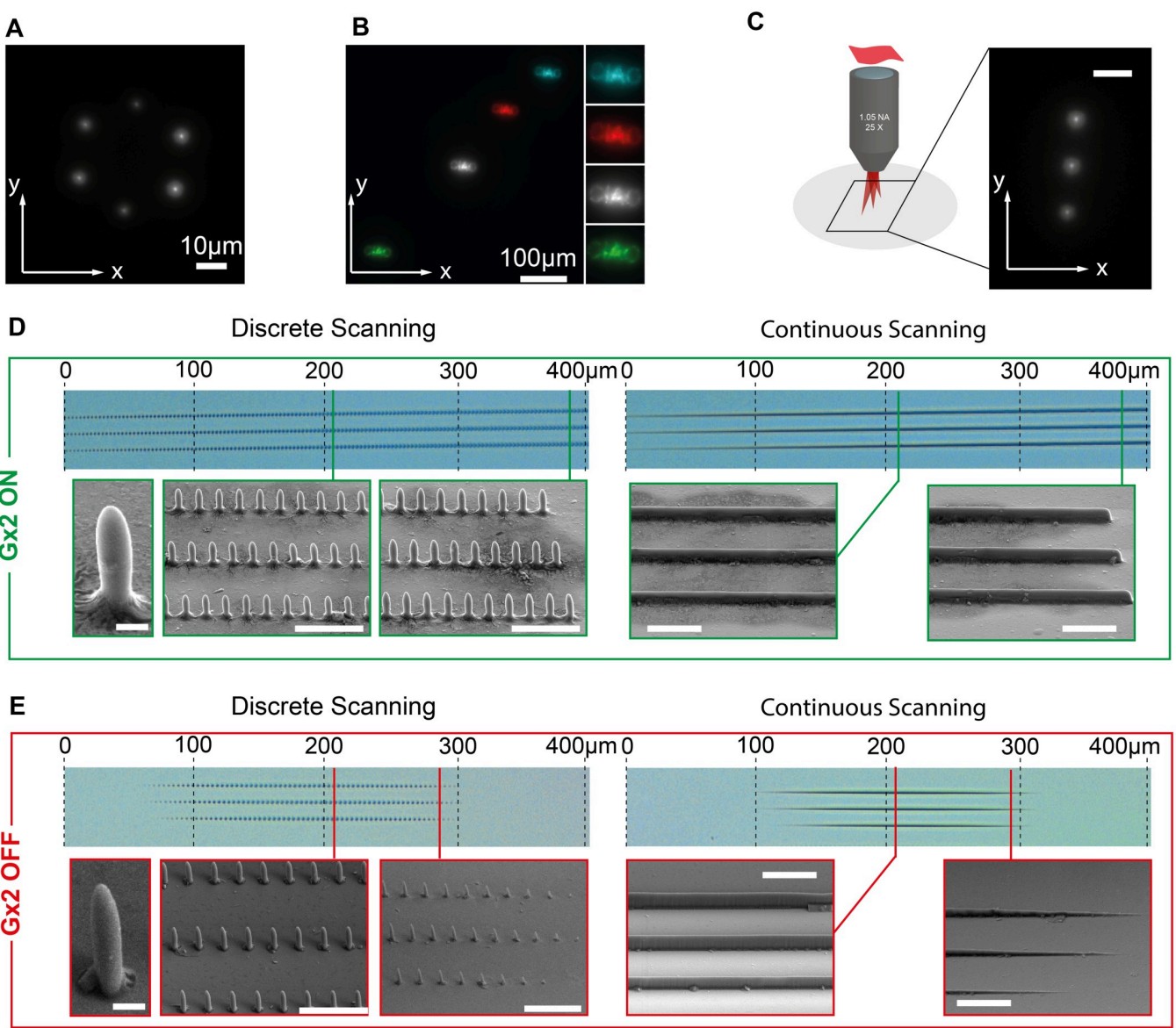

**Fig 4. Holographic projection and polymerization.** (A) Multiple spots generated with the SLM system arranged on the vertex of a hexagon. (B) Complex shape projection on the sample plane shifted in four different positions in the WFOV using the three mirrors scanhead (overlay in false colors of different position). Insets on the right are 2X zooms of the projected patterns. (C) (left) Schematic representation of the writing configuration with the simultaneous projection of three polymerizing spot. The inset on the right shows the fluorescence excited in a drop of fluorescein by the three spots, the scalebar is 10μm. (D) Array of pillars (left) and lines (right) polymerized by discretely and continuously scanning the hologram in panel C with the beam drift compensation implemented by Gx2 enabled. The position of the Scanning Electron Microscope details (bottom rows) are highlighted in the optical microscope image (top) by the vertical lines. Scalebars in the SEM images are 1μm for the single pillar image and 10μm for the others. (E) As in panel (D), without enabling the beam drift compensation implemented by Gx2.

scanheads with bigger entrance pupils and reflective elements are required. This however implies that the distance between the scanning mirrors should increase, making the virtual conjugation paradigm less effective. An alternative solution to the three-mirror scanhead is to separate the two mirrors and create two actually conjugated scanning planes, employing additional scanning optics [24], but this results in increased costs and pulse broadening due to increased group delay dispersion. This latter could be reduced by the use of parabolic mirrors

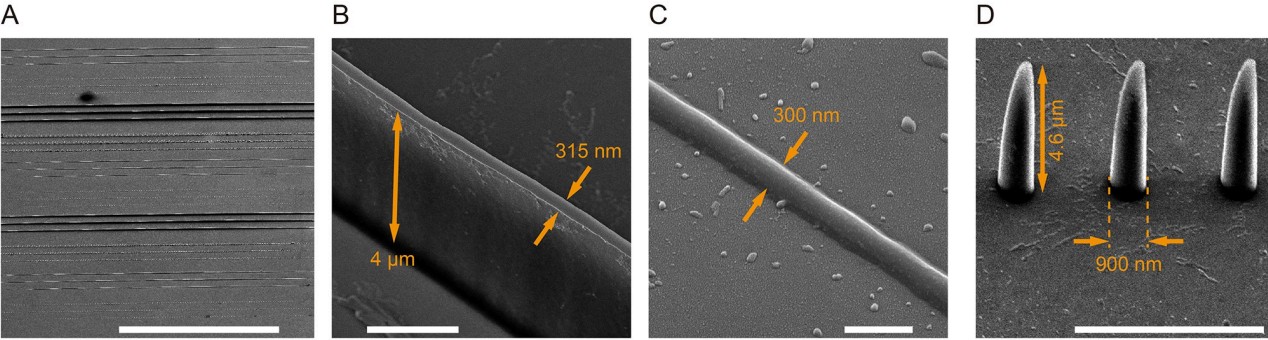

**Fig 5. Structures form factor.** (a) Overall SEM view of array of dots and lines written with three simultaneous polymerizing spots projected holographically at different laser power and dwell time. Scalebar is 200 μm. (b, c) Details of two lines in panel (a) written with laser power/writing speed 59.4mW/10 μm/s (estimated power density of 10.08 TW/cm$^2$ per spot [23]) and 32.6mW/20 μm/s (estimated power density of 5.53 TW/cm$^2$ per spot [23]), respectively; scalebars are 2 μm (panel b) and 500 nm (panel c). (d) Details of dots pattern in (a) written with laser power 59.4mW (estimated power density of 10.08 TW/cm$^2$ per spot [23]) and dwell time 100 ms; scalebar is 5 μm.

as relay element between the galvanometric scanners [25–27], but could also result in increased astigmatism along the optical for any misalignment of the parabolic reflectors [25].

The three-mirrors scanhead instead adds an additional planar reflective surface, and it is effective in writing with a plurality of simultaneously projected polymerizing spots for both discrete and continuous structures with similar shapes with respect to the two-mirror system (see comparison between the shape and size of pillars and lines in Fig 4D and 4E, respectively). A boost of a factor ~2 was obtained in terms of WFOV length in the direction affected by the loss of virtual conjugation, with low additional cost and system complexity with respect to typically employed two-mirror scanhead [1, 11, 16]. A further simplification of the system would rely in the use of a single scanning mirror, avoiding the need of the compensating mirror, which could be particularly useful if patterning is needed preferentially along a single direction. Still, the use of holograms to project and polymerize more complex, solid shapes in the

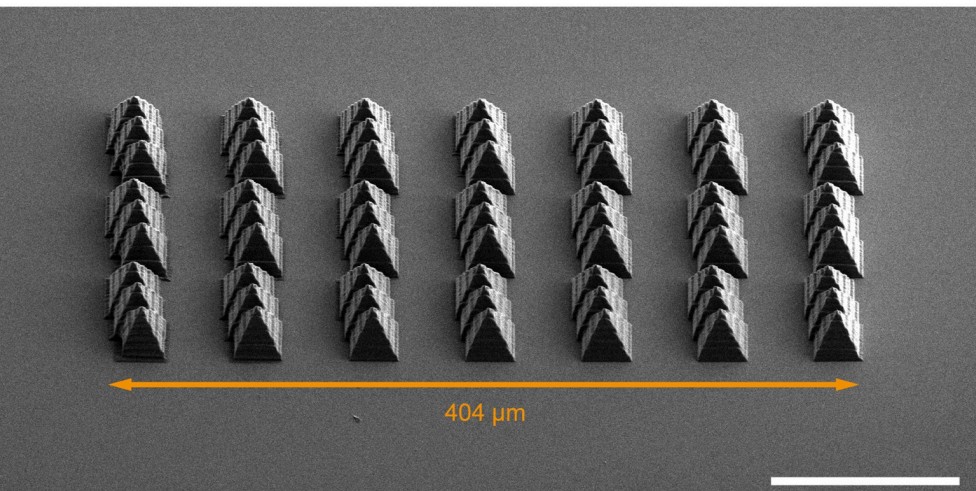

**Fig 6. Writing of 3D structures.** SEM view of an array of pyramids-like structure extending ~400μm along *x* written with three simultaneous polymerizing spot projected holographically with laser power/writing speed 59.4mW/20 μm/s (estimated power density of 10.08TW/cm$^2$ per spot [23]). Scalebar is 100 μm, sample is tilted by 45°.

sample plane, although possible, is limited by the speckle patterns appearing in the reconstructed image (Fig 4B) due both to the coherence of the light source and imperfections in the phase retrieval algorithms used to compute the pattern displayed on the SLM [23]. The resulting unevenness of light intensity is reflected into the polymerized shape. Hardware and software solutions have been proposed so far to reduce the impact of speckles in holographic projection systems, at the expense of the system complexity [23, 28, 29]. The system achieved a rate of 2000 voxels/s when simultaneously writing three 400μm long lines (accounting for voxels overlap to obtain smooth structures), which is smaller than other values reported in literature [30]. This is in part due to the size of the galvanometric mirrors required to host the projected hologram, which reduce the maximum driving speed of the scanhead with respect to other systems [30]. The voxel rate can be improved partly by increasing the number of spots projected simultaneously. As the main aim of this study is exploiting the three mirror scanhead to enlarge the WFOV, this aspect will be investigated in future works.

Notably, the open hardware paradigm allows for straightforward upgrade of the system that could enhance its range of applications: e.g., mounting samples on a closed loop high-resolution three axis translation stage potentially allow for the realization of closely spaced structures in a grating fashion, and for accurate repositioning of the sample during multistep lithographic processes; moreover, hexagonal fields can be easily considered as stitching elements to exploit the higher uniformity of the beam within a circular region. In perspective, we believe our system provides powerful tool that can be exploited for large-scale fabrications of metasurfaces for biosensing [31] and photonics [32, 33], as well as advanced addictive manufacturing [12] with peculiar adaptiveness to unconventional substrates [34].

## Supporting information

**S1 File. Three dimensional CAD model realized with Creo Parametric.** Compressed archive containing the assembly file named "holo2pp3galvos.asm" and all the items files.
(ZIP)

**S2 File. List of all the elements of the system.**
(DOCX)

**S3 File. Custom-written MATLAB software developed to control the system.** Compressed archive containing the software files. Refer to the included readme for further information.
(ZIP)

**S4 File. MATLAB code used to compute an estimation of the relation between the tilts of the mirrors in the scanhead.**
(M)

## Author Contributions

**Conceptualization:** Marco Pisanello, Filippo Pisano, Massimo De Vittorio, Ferruccio Pisanello.

**Data curation:** Di Zheng.

**Funding acquisition:** Massimo De Vittorio, Ferruccio Pisanello.

**Investigation:** Marco Pisanello, Di Zheng, Antonio Balena, Filippo Pisano.

**Project administration:** Ferruccio Pisanello.

**Software:** Marco Pisanello.

**Supervision:** Ferruccio Pisanello.

**Visualization:** Marco Pisanello, Di Zheng, Antonio Balena, Ferruccio Pisanello.

**Writing – original draft:** Marco Pisanello, Di Zheng, Antonio Balena, Ferruccio Pisanello.

**Writing – review & editing:** Marco Pisanello, Di Zheng, Antonio Balena, Filippo Pisano, Massimo De Vittorio, Ferruccio Pisanello.

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
