## [Decision Letter · Decision Letter 0]

19 Nov 2021

PONE-D-21-34208An open source three-mirror laser scanning holographic two-photon lithography systemPLOS ONE

Dear Dr. Pisanello,

Thank you for submitting your manuscript to PLOS ONE. After careful consideration, we feel that it has merit but does not fully meet PLOS ONE’s publication criteria as it currently stands. Therefore, we invite you to submit a revised version of the manuscript that addresses the points raised during the review process.

We look forward to receiving your revised manuscript.

Kind regards,

Maddalena Collini, PhD

Academic Editor

PLOS ONE

Journal Requirements:

"I have read the journal's policy and the authors of this manuscript have the following competing interests: M.D.V. and F.Pisanello are founders and hold private equity in OptogeniX srl, a company that develops, produces and sells technologies to deliver light into the brain. M.D.V.: OptogeniX srl (I). F.P.: OptogeniX srl (I)."  

We note that you received funding from a commercial source: "OptogeniX srl"

Additional Editor Comments:

Dear Authors,

according to the Reviewer comments and to myself, your MS is accepted for publication after the minor points raised by the reviewers have been addressed. In particular, one of the reviewer would be pleased to view the amended MS.

Reviewers' comments:

Reviewer's Responses to Questions

**Comments to the Author**

1. Is the manuscript technically sound, and do the data support the conclusions?

Reviewer #1: Partly

Reviewer #2: Yes

2. Has the statistical analysis been performed appropriately and rigorously? 

Reviewer #1: N/A

Reviewer #2: Yes

3. Have the authors made all data underlying the findings in their manuscript fully available?

Reviewer #1: Yes

Reviewer #2: Yes

4. Is the manuscript presented in an intelligible fashion and written in standard English?

Reviewer #1: Yes

Reviewer #2: Yes

5. Review Comments to the Author

Reviewer #1: The manuscript is relevant for the field of non-linear 3D lithography (laser 3D micro-/nano-printing) and its further developing. The results are original and useful as well as an open source approach is attractive for the scientific and engineering community - it makes it easy to reproduce the revealed findings. Thus in principle the paper can be published in the selected Journal taking into account the indicated questions / remarks:

1) Abstract is not sufficiently specific. For instance the improved writing field by 2 is not compared to initial or the achieved value, nor the absolute value is given. As long as factor of 2 is not disruptive by itself, it should be clarified more technically (2 times corresponding to / in respect to what, or what was the achieved value?). In principle – 2 times is a reasonable improvement in the reported case context. Also, no comments on achieved resolution and throughput is provided, though they both are of critical importance while developing 2PP systems.

2) The obtained manufacturing speeds must be compared to the state-of-the-art achievements. As Authors provide projected (wide field) approach it should be compared to a single beam benchmarking reported results: Optics Express Vol. 27, Issue 11, pp. 15205-15221 (2019) •https://doi.org/10.1364/OE.27.015205. How many individual voxels per second does this novel proposed approach enables to print?

3) Figure 5 has strange units: 4000 nm, but 0.9 um? Why not 4 um and 900 nm?

4) Fig 4: continous => continuous.

5) 690 – 1050 nm < 100 fs laser. Have Authors performed the experiment by varying the wavelength? Seems like not, thus a specific set and applied central wavelength would give more useful information rather than the nominal parameter of the laser which might look misleading.

6) The Authors mention “light intensity” as a parameter resulting to the polymerized shapes, which is indeed correct. Yet the fabrication parameters are provided in mW/um/s – a technical value, which is understandable and perhaps was convenient for the Authors to perform the carried research in the laboratory. However, in a non-linear light-matter interaction case (as is for 2-photon polymerization) it is the intensity (I, [W/cm^2]) the crucial parameter determining the type of interaction and modification of the material. See a recent prominent review paper discussing it: Nanophotonics, vol. 10, no. 4, 2021, pp. 1211-1242. https://doi.org/10.1515/nanoph-2020-0551. Namely Table 2 and Eq. 5 gives a comprehensive definition and explanation for it. So, in order this "open study" to make easier reproducable for other researchers - an intensity at the sample must be calculated. Power units like mW gives little information as different laser sources are used, and if holographic / projection lithography is applied as in the studied case, the multiple beams divide it by some factor (huge, perhaps like 10 or might be 100 - really interesting and important to know for choosing the right laser equipment and is a guide as a requirement for laser engineers). This can be estimated by the Authors and would be useful for the wide spectrum of readers.

7) No real 3D structure is shown! Why? This is the benchmark for the 2-photon polymerization technology – true 3D printing.

Reviewer #2: The Ms. reports an interesting development in the field of the optical setups for fast laser photopolymerization. It specifically tackles with the issue related to the relay between the two galvo mirrors needed for a xy scanning. It is well written and the conclusions are sound and well supported by data. I have only minor points:

1. in the ms. I cannot find a short description of how you get rid of the 0th order of the SLM. In the figure of the setup is marked. Are you adding a Fresnel lens hologram to do that? You can cite literature in this case, for example doi:10.3389/fncel.2014.00092 (2014).

2. in the introduction, on the citation of fabrication on hydrogels (page 2, ine 44), you can also cite other two very recent paper (doi:10.1002/adom.202000584) and review (doi:10.3390/s21175891)

3. Even if it is remarkable that you can increase the FOV with a real xy scanning system cooupled to a SLM, an alternative would be to use one galvo only and the SLM to parallelize the fabrication with a much simpler setup. Could you briefly comment about this possibility, if relevant?

6. PLOS authors have the option to publish the peer review history of their article (what does this mean?). If published, this will include your full peer review and any attached files.

Reviewer #1: **Yes: **Mangirdas Malinauskas

Reviewer #2: No

---

## [Author Response · Author response to Decision Letter 0]

28 Feb 2022

The response to editor and reviewer comments is in the corresponding uploaded document.

---

## [Editor Report · Decision Letter 1]

7 Mar 2022

An open source three-mirror laser scanning holographic two-photon lithography system

PONE-D-21-34208R1

Dear Dr. Pisanello,

We’re pleased to inform you that your manuscript has been judged scientifically suitable for publication and will be formally accepted for publication once it meets all outstanding technical requirements.

Kind regards,

Maddalena Collini, PhD

Academic Editor

PLOS ONE

---

## [Editor Report · Acceptance letter]

7 Apr 2022

PONE-D-21-34208R1 

An open source three-mirror laser scanning holographic two-photon lithography system 

Dear Dr. Pisanello:

I'm pleased to inform you that your manuscript has been deemed suitable for publication in PLOS ONE. Congratulations! Your manuscript is now with our production department. 

Kind regards, 

on behalf of

Dr. Maddalena Collini 

Academic Editor

PLOS ONE